# Shape Directional Asymmetry in Hindlimb Pairs among Calves (*Bos Taurus*)

**DOI:** 10.3390/ani12050559

**Published:** 2022-02-23

**Authors:** Arcesio Salamanca Carreño, Pere M. Parés-Casanova, Oscar Mauricio Vélez Terranova, Néstor Ismael Monroy Ochoa

**Affiliations:** 1Facultad de Medicina Veterinaria y Zootecnia, Universidad Cooperativa de Colombia, Villavicencio 500001, Colombia; nestor.monroy@ucc.edu.co; 2Escola Agrària del Pirineu, Finca de les Colomines, 25712 Bellestar, Spain; pmpares@gencat.cat; 3Facultad de Ciencias Agropecuarias, Universidad Nacional de Colombia, Palmira 763531, Colombia; ovelez@unal.edu.co

**Keywords:** laterality, limb dominance, locomotion, matched symmetry, mediolateral forces, Pyrenean Brown breed

## Abstract

**Simple Summary:**

In biological structures, the shape can be different between the right and left sides. The geometric morphometry technique can contribute to finding these differences by measuring the degree of deviation known as asymmetry. Deviations can occur in response to environmental stress, biomechanical pressures, genetic effects or lateralized behavior. The aim of this study was to determine the degree of asymmetries in the autopods of the hind limbs in healthy Brown Pyrenean calves, using the geometric morphometry (GM) technique. In a slaughterhouse, post-mortem samples of 28 autopods (right–left) were taken from the hind limbs. Dorsoplantar radiographs were obtained for each autopod, and then the bone shape was compared in right–left pairs using 15 landmarks. The results show that the right and left distal extremities present a certain degree of symmetry, but they develop differently in direction; the right hind limbs tend to rotate outwards, and the left hind limbs tend to rotate inwards. The results stimulate the evaluation of the function of each hind limb of cattle during standing and locomotion. This study can be considered as the first symmetric structural investigation of this cattle using geometric morphometry.

**Abstract:**

The aim of this study was to determine paired asymmetries (right–left) in the autopods of bovine hindlimbs using geometric morphometry (GM). A total of 28 hindlimb right–left matched autopods belonging to healthy Brown Pyrenean calves were assessed. Dorsoplantar radiographs were obtained for each autopod. The bone shape was compared on right and left pairs by means of GM techniques, using a set of 15 landmarks. The results suggest that right and left distal limbs are, despite a perceived resemblance of symmetry, differently directionally developed in shape, with right hindlimbs tending to supinate (rotate outwards) and left hindlimbs tending to pronate (rotate inwards). This unevenness is probably related to the mediolateral forces’ contribution of each limb in carrying out the tasks of propulsion and control during walking, and/or a consequence of a laterality associated with a lateralized grazing posture. Our findings prompt a new reassessment of the function of each bovine hindlimb during standing and locomotion.

## 1. Introduction

Form in bilateral structures can differ between the right and left sides [1,2]. These asymmetries are expressed, normally, as fluctuating asymmetry or as directional asymmetry [3,4,5]. Fluctuating asymmetry (FA) represents minor non-directional deviations from symmetry [3]. FA is the response to genetic and environmental stress reflecting the degree of adaptation of the organism [6], and it is considered a negative indicator of the ability to resist small developmental disturbances [3]. FA has also been used for the evaluation of phenotypic variability within populations and its relation to genetic variability and geographic isolation [7].

Directional asymmetry (DA) occurs when there is a meaningful directional deviation [8,9] and, among other reasons, can be the result of a lateralized behavior and biomechanical pressures [10]. DA can result from genetic inheritance, or from the functionality that certain traits acquire in the environment in which they develop [11].

It has been generally assumed that, among cattle, limbs are symmetrical. However, confirmative research on this topic is very scarce [12,13], and few studies have been based on geometric morphometrics [14,15]. The aim of this study was to determine matched (right–left pair) asymmetries in bovine hindlimbs and, more specifically, in the autopods, assessed by geometric morphometrics (GM) on radiographs. The GM technique is based on Cartesian coordinates of landmarks, e.g., anatomical points which are homologous across specimens [16], and on which size, position and orientation have been eliminated from the mathematical shape [16,17]. One of the interests of our study is not only its methodological novelty, but also its implications on limb evaluation of cattle maintained under semi-extensive conditions (outdoor management), as the conclusions can be viewed as the expression of a “naturally balanced” tendency. It is for this reason that in animals kept in conditions of semi-extensiveness, where biomechanical functionality remains little altered, their study could reflect the “normal” behavior of these functional deviations.

## 2. Materials and Methods

### 2.1. Sample

A post-mortem sample of 28 pairs of hindlimbs (autopods) from Pyrenean Brown calves (<14 months) randomly selected in a commercial slaughterhouse was collected. The Pyrenean Brown is a local cattle breed raised mainly in the Catalan Pyrenees of Northeastern Spain, which is managed under semi-extensive conditions (grazing all year round) and only for meat production. Sampled individuals presented no limpness, swelling or joint distension at the sacrifice moment. None of them had been trimmed. Sex and carcass weight were not considered, although in the sample, there were no castrated cattle. Following sacrifice at the slaughterhouse, sectioning of the limbs at the tarsal level was carried out. Limbs were collected after this sectioning occurred, cleaned extensively in running water, dried and kept in containers until being transported to the laboratory. Approval for the present study was not required as animals had been killed for commercial purposes unrelated to this research.

### 2.2. Imaging

Radiographic images were obtained at our laboratory using a high-resolution digital system. Limbs to be examined were placed in a natural stand on a block, and the X-ray beam was centered approximately on the fetlock. Exposure factors were 60 kV and 3.2 mAs. Captured images were then transformed to TpsUtil v.1.40 software (NY, USA) [18], and landmarks were recorded using TpsDig v.2.26 software [19] twice in two independent sessions. A total set of 15 landmarks on the acropodium was used on each image (Figure 1). Their position was based on previous works [15].

### 2.3. Statistical Analyses

Firstly, we used TpsSmall v.1.33 software [18] to test whether the observed variation in shape was sufficiently small so that the distribution of points could be used as a good approximation of the shape space [20]. The correlation between the tangent space *Y* regressed onto the Procrustes distance was 0.9995, so there was a very good approximation of the shape space by the tangent space. This made ulterior estimates of shape differences reliable. To obtain the shape data, landmark configurations were then superimposed using the generalized Procrustes method, based on the minimization of the distance between corresponding landmarks [20], by translating, rotating and scaling all configurations [21]. As a proxy for size, we used the centroid size, which corresponds to the “squared root of the sum of the squared distances from each landmark to the centroid” [20]. The centroid size contains information about the actual size prior to superimposition [22]. To test the significance of allometry, a regression of asymmetric shape scores vs. centroid size was performed. Then, we assessed superimposed landmarks for left–right matched asymmetries of form in relation to individuals, sides (DA), their interaction (FA) and measurement error. Finally, a thin plate spline (colored representation of bending energy) allowed us to appreciate shape changes as forces acting on a mesh that deforms according to the sense, showing expansion or contraction on each landmark.

All analyses were processed customly through MorphoJ v.1.07a (Manchester, UK) [23] and PAST v.1.06c (Oslo, Norway) [24]. The confidence level was established at 95%.

## 3. Results

Measurement errors were considered negligible for shape, accounting for a mere 2.3% of the total observed variance (Table 1). Procrustes ANOVA showed highly significant variations in symmetry within the “individual × side” interaction (FA) and “side” effects (DA), the former being much higher than the latter (71.9% vs. 6.9%) (Table 1). The observed shape asymmetry was associated with size changes, i.e., differences were allometric (*p* = 0.0019; 10,000 randomization rounds), with 5.9% of the shape change explained by the size change, and thus a final thin plate spline was conducted on regression scores. According to this representation, shape changes affect the most distal part, with a clear lateral (right) displacement of the right autopod and an inward displacement of the left autopod. In this thin plate spline, yellow and red areas represent expansion, while blue areas represent contraction (Figure 2).

## 4. Discussion

Traditionally, it has been assumed that the two limbs of artiodactyls are exactly equal [25]. Here, we present a radiographic investigation of matching symmetry in autopods, demonstrating that right and left hindlimb pairs are differently directionally developed in shape, in healthy young bovines. Thus, there are significant differences between pairs, despite a perceived resemblance of symmetry. Matched asymmetry appears to be the norm, with different right–left autopodium orientations. Fluctuating asymmetry seems not to be a consequence of developmental instability and merely superimposed on directional asymmetry.

Having carried out this study on animals kept in conditions of semi-extensiveness, where biomechanical functionality remains little altered, it reflects that this asymmetry represents a “normal” anatomical deflection. These bilateral differences may be explained by differences in limb dominance, e.g., caused by a different level of activity (force generation and/or foot posture) for right or left hindlimbs. Each limb would not restrict its movement to the sagittal plane during straight forward movement but would be tuned to different forces for stability during stance, producing medial or lateral pull forces [26] differently on each side and therefore indirectly causing unevenness. This unevenness may be a consequence of laterality associated with a lateralized grazing posture, too, which has been described in horses [27]. Another possible explanation for this asymmetry could be viewed from a deflection pattern, as ruminants carry a huge, heavy rumen to the left, with further load asymmetry stemming from a fluctuating uterus to the right.

Whether and to what degree these differences are accompanied by corresponding differences in soft tissue structures and function require further study, for instance, by ultrasonography. Equally, future studies are needed to evaluate asymmetries at other ages and management, and to determine if the motion produced by the hindlimbs results in similar energy conservation. Anyway, our current findings prompt a reassessment of the function of each of the limbs during standing and locomotion. In addition, the results of this type of analysis will be improved by their application to 3D conformations, in this way avoiding possible complications derived from the analysis of 2D images.

## 5. Conclusions

The results show significant differences in the shape of matched distal hindlimbs of domestic cattle, at least in the studied sample belonging to calves managed semi-extensively. This research, to the best of our knowledge, presents the first such symmetric structural investigation in cattle using geometric morphometrics.

## Figures and Tables

**Figure 1 animals-12-00559-f001:**
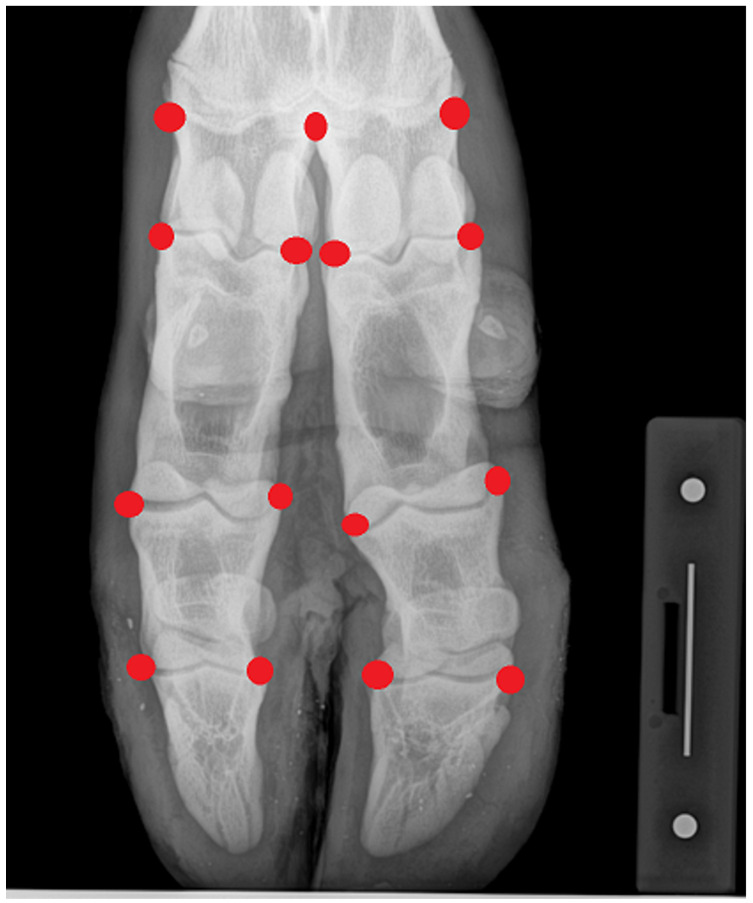
Dorsoplantar view of bovine autopod (in natural stand) on which 15 landmarks occurring on the acropodium for each limb were located.

**Figure 2 animals-12-00559-f002:**
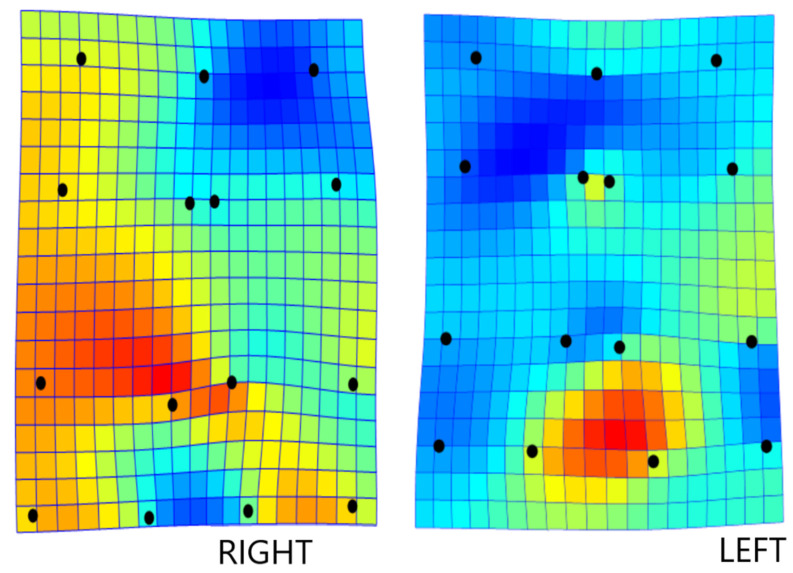
Representation of bending energy (thin plate spline), which allows shape changes to be appreciated as forces acting on a colored mesh. This interaction deforms according to the direction of right and left matched hindlimb autopods (*n* = 28 pairs). Dorsal view. Shape changes mainly affect the distal part—with a clear expansion (displaced outwards), reddish coloration, on the right autopod and an inward displacement of the left autopod.

**Table 1 animals-12-00559-t001:** Procrustes ANOVA for shape of hindlimbs of Pyrenean Brown calves (*n* = 28 pairs). The individual’s effect denoted the individual variations in shape. MS (mean square) is the sum of squares divided by the appropriate degrees of freedom, reflecting the magnitude of the effect. The main effect of “side” indicated the variation between sides and was considered as the measure of directional asymmetry. The “individuals × side” effect is the mixed effect which indicates there is fluctuating asymmetry.

Effect	SS	MS	Df	F	P
Individual	0.03927195	0.0001258716	312	2.69	<0.0001
Side	0.01256665	0.0004833328	26	10.35	<0.0001
Individual × side	0.01457452	0.0000467132	312	2.97	<0.0001
Error	0.01064547	0.0000157477	676		

## Data Availability

Data are available upon request to the second author.

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
