# Peer review of "Shape Directional Asymmetry in Hindlimb Pairs among Calves (Bos Taurus)"

_animals, 2022, doi:10.3390/ani12050559_

Round 1

Reviewer 1 Report

In this manuscript, the authors present a study to determine paired asymmetries in the autopods of brown pyrenean calves throughout Geometric Morphometry. The relevance of this research it is in the the lack of publications and evidences on this type alterations. In this sense, the authors provide a new rich information about the topic. For these reasons, the publication of this work is recommended, as a starting point for the publication of similar evidences of other taxa and management contexts.

However, before that, some issues to be corrected. In general, the information is expressed correctly, but punctuation marks must be corrected, as very long sentences are created. Images of the original material are lacking, as well as a broader explanation of the relationship of the identified asymmetry and the type of livestock management.

Below, I add a list of comments and questions to be corrected in the text. I make some brief comments on issues to improve in different sections.

Manuscript:

  • Line 13: better “degree of deviation known…”
  • Line 14-16: use “,” after The aim of the study… and after Brown Pyrenean calves, using…
  • Line 18: use “,” after autopod, and then, the…
  • Line 19: use “,” after degree of symmetry, but…
  • Line 26: delete “was sampled post-mortem from an abattoir”. Explain this in the materials and methods.
  • Line 27: better “…each autopode. The bone shape was compared on right and left pairs by meand of GM techniques, using a set of 15 landmarks”.
  • Line 30: better “developed in shape. Right hindlimbs…”.
  • Line 46: use “,” after withn populatins, and its relation…
  • Line 66: delete “by first author”.

Comments:

  • A more detailed explanation of how this type of asymmetry can be produced by semi-extensive livestock is needed. Both in the discussion and in the introduction.
  • Review the use of punctuation marks throughout the manuscript.
  • Include figure with the original sample or a selection of them in the “Materials and methods” section.
  • Reorder captions for figures and tables.

Author Response

First reviewer’s 1 responses

Comment

Line 13: better “degree of deviation known…”

Response: corrected

Comment

Line 14-16: use “,” after The aim of the study… and after Brown Pyrenean calves, using…

Response: added second “,” (we do not understand the use of “,” after The aim of the study…)

Comment

Line 18: use “,” after autopod, and then, the…

Response: added

Comment

Line 19: use “,” after degree of symmetry, but…

Response: added

Comment

Line 26: delete “was sampled post-mortem from an abattoir”. Explain this in the materials and methods.

Response: deleted

Comment

Line 27: better “…each autopode. The bone shape was compared on right and left pairs by meand of GM techniques, using a set of 15 landmarks”.

Response: corrected

Comment

Line 30: better “developed in shape. Right hindlimbs…”.

Response: corrected

Comment

Line 46: use “,” after withn populatins, and its relation…

Response: corrected

Comment

Line 66: delete “by first author”.

Response: deleted

Comment

A more detailed explanation of how this type of asymmetry can be produced by semi-extensive livestock is needed. Both in the discussion and in the introduction.

Response:  the following sentence has been added: “It is for this reason that in animals kept in conditions of semi-extensiveness, where biomechanical functionality remains little altered, their study could reflect the "normal" behavior of these functional deviations.” (lines 61-63) and “Having carried out this study in animals kept in conditions of semi-extensiveness, where biomechanical functionality remains little altered, would reflect that this asymmetry represents a "normal" anatomical deflection.” (lines 141-143). We think that they clarify the relationship of the identified asymmetry and the type of livestock management.

Comment

Review the use of punctuation marks throughout the manuscript.

Response: done, and long sentences have been shortened

Comment

Include figure with the original sample or a selection of them in the “Materials and methods” section.

Response: we do not understand this comment. Moreover, which image of original material is suggested to be included?

Comment

Reorder captions for figures and tables.

Response: we do not understand this comment

Reviewer 2 Report

First, let me point here to the most important flaw from my view point: while hinting that everything short of coriolis may distort the limbs, you fail to make any reference to the fact that they are ruminants and thus carry along a huge, heavy rumen to the left, even in the absence of data from multiparous females, with further load asymmetry stemming from a fluctuating uterus to the right.

Then, your english is quite wrong even for my non-maternally trained ears: so many missing articles, frequently wrong use of verbal persons, that (line 13) very hurting "or not", (line 23) "of this cattle", an unknown countable item in line 53, line 58 where you mean "removed" but say "excluded" for great confussion,...

Regarding your Methods section, I deprecate the elusion of details from significant elements of your methodology through plain reference to the software tools, which indeed are not where the references point. As for the Sample section, it may be of interest for some readers that you describe in more detail the sample collection procedure and particullarly the choice of landmarks, for which I very much miss the apex of those autopodes.

Finally, would you please extend on that most seminal conclusion in line 138?

Author Response

Second reviewer’s responses

Comment

First, let me point here to the most important flaw from my view point: while hinting that everything short of coriolis may distort the limbs, you fail to make any reference to the fact that they are ruminants and thus carry along a huge, heavy rumen to the left, even in the absence of data from multiparous females, with further load asymmetry stemming from a fluctuating uterus

to the right.

Response: Very interesting comment. Effectively, the Coriolis effect describes the pattern of deflection, but taken by objects not firmly connected to the ground. But the hypothesis of assymetrically positioned viscera have been commented: We have added the following sentence: “Another possible explanation of this asymmetry could be vew from a deflection pattern, as ruminants carry along a huge, heavy rumen to the left, with further load asymmetry stemming from a fluctuating uterus to the right.” (lines 150-153).

Comment

Then, your English is quite wrong even for my non-maternally trained ears: so many missing articles, frequently wrong use of verbal persons, that (line 13) very hurting "or not", (line 23) "of this cattle", an unknown countable item in line 53, line 58 where you mean "removed" but say "excluded" for great confusion...

Response: Corrected. The English wording in the manuscript has been revised. Changes appear in blue

Comment

Regarding your Methods section, I deprecate the elusion of details from significant elements of your methodology through plain reference to the software tools, which indeed are not where the references point. As for the Sample section, it may be of interest for some readers that you describe in more detail the sample collection procedure and particularly the choice of landmarks, for which I very much miss the apex of those autopodes.

Response: a new sentence has been added (line 84): “Their position was based on previous works [15]”.

Comment

Finally, would you please extend on that most seminal conclusion in line 138?

Response: an entire sentence has been deleted as it creates, in our opinion, a certain confusion.
